# Neuroinductive properties of mGDNF depend on the producer, *E. Coli* or human cells

**Dzhirgala V. Shamadykova**[1,2]*, **Dmitry Y. Panteleev**[1], **Nadezhda N. Kust**[2], **Ekaterina A. Savchenko**[3], **Ekaterina Y. Rybalkina**[4], **Alexander V. Revishchin**[1], **Galina V. Pavlova**[1,3,5]*

**1** Institute of Higher Nervous Activity and Neurophysiology, Russian Academy of Sciences, Moscow, Russia, **2** Institute of Gene Biology, Russian Academy of Sciences, Moscow, Russia, **3** Burdenko Neurosurgical Institute, Moscow, Russia, **4** Blokhin Russian Cancer Research Center, Moscow, Russia, **5** Institute of Molecular Medicine, Sechenov First Moscow State Medical University, Moscow, Russia

* djirgala04@gmail.com (DVS); lkorochkin@mail.ru (GVP)

## Abstract

The glial cell line-derived neurotrophic factor (GDNF) is involved in the survival of dopaminergic neurons. Besides, GDNF can also induce axonal growth and creation of new functional synapses. GDNF potential is promising for translation to treat diseases associated with neuronal death: neurodegenerative disorders, ischemic stroke, and cerebral or spinal cord damages. Unproductive clinical trials of GDNF for Parkinson's disease treatment have induced to study this failure. A reason could be due to irrelevant producer cells that cannot perform the required post-translational modifications. The biological activity of recombinant mGDNF produced by *E. coli* have been compared with mGDNF produced by human cells HEK293. mGDNF variants were tested with PC12 cells, rat embryonic spinal ganglion cells, and SH-SY5Y human neuroblastoma cells *in vitro* as well as with a mouse model of the Parkinson's disease *in vivo*. Both *in vitro* and *in vivo* the best neuro-inductive ability belongs to mGDNF produced by HEK293 cells. Keywords: GDNF, neural differentiation, bacterial and mammalian expression systems, cell cultures, model of Parkinson's disease.

**Data Availability Statement:** All relevant data are within the manuscript and its Supporting information files. All original image data are also available from the Harvard Dataverse Network

## 1. Introduction

To date, neurotrophic factors are the most potent mediators of neuronal survival and inducers of their differentiation. The glial cell-derived neurotrophic factor (GDNF) is given particular attention among neurotrophic factors. GDNF was identified in the medium conditioned by a glial cell line based on the capacity to promote neuronal survival as well as cell enlargement and neurite elongation in mesencephalic dopaminergic neurons *in vitro* [1]. GDNF was initially considered as a selective survival factor of nigrostriatal dopaminergic neurons [2]. An exogenous infusion of GDNF into the striatum in a mouse model of Parkinson's disease prevented the degeneration of A9 dopaminergic neurons in the substantia nigra, which points to the retrograde transport of GDNF [3,4]. Subsequent studies demonstrated that GDNF also sustained the survival of spinal motoneurons [5] and stimulated axonal growth in a culture of hippocampal neurons from a 19-day rat embryo [6] as well as stimulated the survival, migration,

database (accession number https://doi.org/10.7910/DVN/DIYBUS).

**Funding:** G.P. received grant funding. This research was funded by the Ministry of Science and Higher Education of the Russian Federation (grant number 075-15-2020-809 (13.1902.21.0030) and RFBR (grant number 18-29-01-012). The funders had no role in study design, data collection and analysis, decision to publish, or preparation of the manuscript.

**Competing interests:** The authors have declared that no competing interests exist.

**Abbreviations:** cmGDNF, mGDNF isolated from the conditioned medium (HEK293-derived; GDNF, glial cell line-derived neurotrophic factor; mGDNF, mature GDNF; MPP+, 1-methyl-4-phenylpyridinium ion; MPTP, 1-methyl-4-phenyl-1,2,3,6-tetrahydro-pyridine; MTT, 3-(4,5-Dimethylthiazol-2-yl)-2,5-diphenyltetrazolium bromide; SNpc, substantia nigra pars compacta; TH, tyrosine hydroxylase; VTA, ventral tegmental area.

and differentiation of certain peripheral neurons [7]. GDNF also maintains motor, sympathetic, parasympathetic, sensory, and enteral neurons [8].

Experiments on rodents and primates demonstrated the efficiency of GDNF as a potential therapeutic agent; however, clinical trials in humans proved controversial [9]. For example, Barry Hoffer and later Anders Björklund with colleagues experimentally studied GDNF as an agent restoring dopamine neurons in animal models of Parkinson's disease [4,10,11] and obtained quite promising results. However, during the Phase II of clinical trials no efficiency of recombinant GDNF had been found, and that was the reason of underestimation of possibility of its application for this therapeutic area. Several attempts were made to explain the unsuccessful results. For instance, recombinant GDNF could have abnormal distribution in the cerebral parenchyma due to its heparin-binding properties [12] or could be immunogenic [13] Potential causes were discussed by Tenenbaum and Humbert-Claude [14].

Further modifications of the methods and doses of factor administration can lead to more stable positive clinical trials [9,14,15]. Other approaches to efficient GDNF application can include molecular modifications of the recombinant factor and generation of transgenic producer cultures [16,17].

The last approaches are the most perspective and they require further study. The reason of unsuccessful clinical trials could be, for example, a wrong isoform of the protein selected for the therapy, because of diversity of natural GDNF. Intensive studies of GDNF demonstrated a variety of isoforms with specific secretion as well as a variety of receptor complexes influenced by the factor and patterns of such influence.

Two GDNF isoforms encoded by *pre-(α)pro-GDNF* and *pre-(β)pro-GDNF* differ by the pro-region length and secretory pathway [18]. The endogenous protein is expressed as a mature GDNF (mGDNF) or its precursor (pro-GDNF). The signaling peptide is cleaved early in protein translation [18]. The mature GDNF is a glycosylated protein with the molecular weight of ~25 kDa composed of 134 amino acids, and it functions as a homodimer [19].

The deletion of the pre- and pro-regions of the *GDNF* gene does not interfere with the secretion of its transgenic products from mammalian cells (HEK293) [20]. Simultaneous removal of the pre- and pro-regions enhanced the trophic effect of the mature factor. A neuroinductive effect on mouse neural precursor cells (embryonic spinal ganglia) was demonstrated for mGDNF/GFP obtained from transgenic HEK293 cells [20]. The capacity of mGDNF/GFP synthesized in transgenic mammalian HEK293 cells to induce neural differentiation was demonstrated using the classical *in vitro* model with PC12 cells [21]. After transplantation of transgenic cells producing mGDNF/GFP into the brain of mice with chemically induced Parkinson's disease (1-methyl-4-phenyl-1,2,3,6-tetrahydro-pyridine (MPTP) administration), mGDNF/GFP release significantly increased the number of dopaminergic neurons in the substantia nigra, which restored the mouse motor activity. Notice that the mGDNF/GFP from mammalian cells, specifically, human embryonic kidney HEK293, was used in these experiments, which has both advantages and disadvantages. Besides, it is important to understand whether *E. coli* is applicable for producing of mGDNF, and what would be properties of the protein after changing the expression system as well as removal of GFP fusion region for the C-terminal. Recombinant protein production always raises the question of the changes dependent on the expression systems and the deletion of the C-terminal GFP fusion sequence. To date, several techniques for the optimal expression of recombinant proteins are available [22,23] based on different cell systems. The cellular expression systems producing transgenic proteins can be divided into prokaryotic and eukaryotic [24]. Prokaryotic systems are more popular; they provide a high yield among other advantages [24–27]. Most recombinant proteins used in experiments were generated using this approach. The technical simplicity of mass production of recombinant proteins made *E. coli* the most popular prokaryotic system of

protein expression [28]. Despite clear advantages, this system has drawbacks including abnormal post-translational modifications, improper protein folding, and accumulation of the recombinant protein in the inclusion bodies, which complicates its isolation and refolding [25]. Apart from bacteria, yeast and insect cell systems are used to produce recombinant proteins [29–31]. In contrast to *E. coli* cells with a high growth rate and product yield, yeasts provide for post-translational modifications.

Widely used eukaryotic cell systems include mouse fibroblast cells (C 127-BPV), Chinese hamster ovary cells (CHO-DHFR, CHO-NEOSPLA, CHO-GS), and mouse myeloma cells (NSO-GS) [32,33]. There is also a trend of using human cell lines to produce human proteins [30]. One of the advantages of human cell lines is the proper post-translational modifications corresponding to endogenous human proteins [34]. Recent data indicate different glycoprotein profile of the protein expressed in cells of different kingdoms (prokaryotes and eukaryotes) as well as cells of different species and lines within the same kingdom [35–37]. Such differences can skew the results and lead to conclusions that do not correspond to the real *in vivo* processes.

Here, the neuroinductive efficiencies of mGDNF synthesized in prokaryotic (*E. coli*) and eukaryotic (human HEK293) producers were compared. The biological activity of the obtained proteins was analyzed *in vitro* using the PC12 cell line, rat spinal ganglion cells, and SH-SY5Y cell line. In addition, the neuroprotective activityof mGDNF (produce from *E. coli* or HEK293) was studied in the MPTP model of Parkinson's disease *in vivo*, the tyrosine hydroxylase activity of substantia nigra pars compacta was assessed, and the motor activities of mice, treated with GDNF isoforms, was compared. The main goal of the research is to understand how important to produce mGDNF in mammalian cells for the keeping mGDNF functioning, or whether it is possible to produce it from *E. coli.*

## 2. Materials and methods

### 2.1 Cloning

The nucleotide sequences of the modified *GDNF* gene (Gene ID:2668), *mGDNF*, *Pre-mGDNF*, and *Pro-mGDNF*) were amplified using specific primers. NdeI and BamHI restriction sites were introduced into these modifications.

The following primers were used:

*Gdnf(m)NdeI (F)*, 5'-CATATGTCACCAGATAAACAA-3';

*NdePrePet15 (F)*, 5'-CATATGAAGTTATGGGATGTCG-3';

*NdeProPet15(F)*, 5'-CATATGTTCCCGCTGCCCGCCG-3';

*GdnfstpBamHI (R)*, 5'-TGGATCCTCAGATACATCCACACCTTTTAGCG-3'.

The PCR amplification program consisted of 30 cycles: 90 s at 94°C, 20 s at 57–62°C (depending on primers), and 15 s at 72°C, with a final extension of 10 min at 72°C. The PCR products were analyzed on a 1.5% agarose gel containing ethidium bromide. The following PCR products were synthesized: *mGDNFstp* (405 bp), *Pre-mGDNFstp* (462 bp), and *Pro-mGDNFstp* (579 bp). The amplification products were cloned into pGEM-T Easy (Promega, USA). The recombinant clones were selected by the blue/white screening. As a result, three intermediate constructs were generated: pGEM/*mGDNF*stp, pGEM/*Pre-mGDNF*stp, and pGEM/Pro-mGDNFstp. The constructs were verified by restriction analysis and subsequent sequencing using the universal M13 reverse and M13 forward primers. The NdeI-BamHI fragments were cloned into the corresponding sites of pET15b (5708 bp), which yielded the following constructs: mGDNFstp/pET15b, Pre-mGDNFstp/pET15b, and Pro-mGDNFstp/pET15b.

## 2.2 Transformation of the obtained constructs into *E. coli* to obtain producers of the modified factors

The constructs were transformed into *E. coli* strain BL21 (DE3) synthesizing T7 RNA polymerase. The colonies selected on LB-ampicillin plates of Rosetta(DE3)/mGDNFstp/pET15b, Rosetta(DE3)/Pre-mGDNFstp/pET15b, and Rosetta(DE3)/Pro-mGDNFstp producers were replated on LB-ampicillin plates. The resulting transformed *E. coli* BL21 (DE3) cells with *mGDNF*stp/pET15b, *Pre-mGDNF*stp/pET15b, or *Pro-mGDNF*stp/pET15b were maintained as a streak culture.

## 2.3 Expression of recombinant GDNF genes (mGDNF, Pre-mGDNF, and Pro-mGDNF) protein purification

*E. coli* cells (strain BL21 (DE3) expressing T7 RNA polymerase) transformed with one of GDNF modifications, mGDNF, Pre-mGDNF, or Pro-mGDNF, were grown in LB-ampicillin with agitation at 25–37˚C: mGDNF- and Pre-mGDNF-transformed cells, at 25˚C; Pro-mGDNF, at 37˚C. The expression was induced by 0.5 mM isopropylthiogalactoside and cells were cultured for 4 h. *E. coli* cells were harvested by centrifugation at 4000g for 15–20 min. The culture broth of the producers of recombinant mGDNF (750 mL) was cooled and the biomass was harvested by centrifugation at 9000*g* and 4˚C for 15 min. The pellet was washed once with 5 mL of PBS, pH 8.0, and ultrasonicated on ice for 1 min. The suspension was centrifuged at 18000 g and 4˚C for 15 min. The pellet was subjected to such ultrasonication and centrifugation three times more.

The pellet containing inclusion bodies with the recombinant proteins was dissolved in 5 mL of buffer containing 8 M urea, 0.01 M imidazole, 0.05 M DTT, and 0.05 M Tris-HCl, pH 10, by ultrasonic disintegration, and centrifuged. The supernatants were filtered through a 0.22 µM membrane and applied onto a 1.5×5 cm column with Ni Sepharose 6 Fast Flow (GE Healthcare) equilibrated with the above buffer at room temperature equipped with an adapter for a peristaltic pump and a UV detector at 280 nm. The column was washed with the same buffer and eluted with buffer containing 8 M urea, 0.25 M imidazole, 0.05 M DTT, and 0.05 M Tris-HCl, pH 10, until complete elution from the column. The resulting effluents were analyzed by 10% PAGE (Fig 1) and stored at -20˚C until dialysis, refolding, and concentration of the protein.

The total yield of the purified protein was 4–5 mg/l for mGDNF, 1 mg/mL for Pro-mGDNF or 0,2 mg/mL for Pre-mGDNF. Protein concentration was determined by the Bradford assay.

Proteins were separated by electrophoresis in 10% acrylamide gel. The 8–2 M urea gradient dialysis of proteins was performed at room temperature in sacks with a pore size of 14 kDa and a flow rate of 0.1 mL/min (PBS with 0.5 M arginine for mGDNF or PBS with 0.5% Triton X100 and 0.5 M arginine for Pro-mGDNF). As a result, a lossless recovery of mGDNF and Pro-mGDNF was achieved. To solve the identified problem of the recombinant protein aggregation, the final conditions were refined by the dialysis of eluates with the minimal concentrations of recombinant proteins using the Pierce Microdialysis Plate against 100mM Tris-HCl, pH 9 and 10, and 5, 20, and 100mM DTT. Thus, two GDNF modifications have been obtained: mGDNF and Pro-mGDNF.

A folding modification of the recombinant mGDNF protein was performed by the dialysis of the protein in the presence of animal recipient serum (referred to as mGDNF2-Coli). The eluate was supplemented with an equal volume of the mouse serum and dialyzed using a membrane with an MW cutoff of 12 kDa at room temperature with agitation for 16 h against a

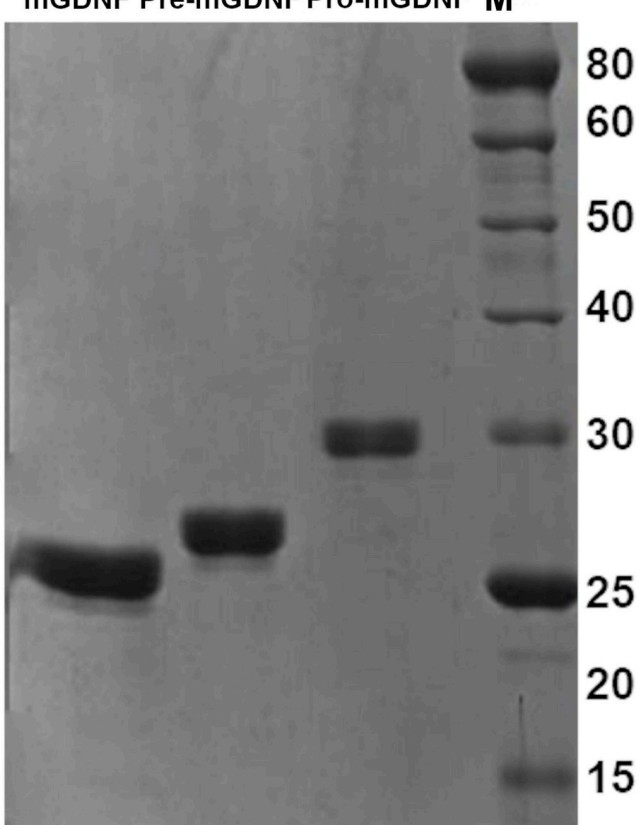

**Fig 1. Electrophoretic mobility of GDNF proteins: mGDNF (1), Pre-mGDNF(2), and Pro-mGDNF (3).** 10% PAGE of proteins purified by chelate chromatography on NiNta columns in 8 M urea after dialysis against PBS.

1000× volume of 150 mM NaCl and 10 mM Tris-HCl. Aliquots of the purified protein were frozen in liquid nitrogen and stored at -70˚C.

## 2.4 Isolation of cmGDNF

The mGDNF protein was isolated from the medium conditioned by HEK293 cells expressing mGDNF/GFP by immunoprecipitation using GFP-Trap agarose (Chromotek, Germany). The conditioned serum-free medium after 48-h incubation with transformed HEK293 cells (10 mL) was filtered through a 0.22 μm membrane and incubated with 30 μl of washed agarose at 4˚C for 2 h. After immunoprecipitation, agarose was sedimented by centrifugation (1000 g, 3 min) and washed with 2 mL of cold buffer containing 10mM Tris-HCl, 150mM NaCl, and 1mM EDTA, pH 8.0, and then with high-salt buffer containing 300 mM NaCl. Before elution with 50 μl of 200 mM glycine (pH 2.5), agarose was washed once with 1 mL of 5 mM Tris-HCl, pH 7.5. Tris-HCl (1M, pH 7.5, 1 μl) was preliminary added to the tube where the final eluate was collected. The protein concentration in the resulting solution assayed using a NanoDrop 2000 spectrophotometer at 280 nm was 200 μg/mL. The obtained aliquots (referred to as cmGDNF) were stored at -70˚C.

Recombinant mGDNF/Coli (produced in-house) was taken as internal control of protein production under the same conditions, recombinant commercial recGDNF (Peprotech, USA)

was chosen as an external control, as a conventional control accepted worldwide for studies of neutrophic properties of GDNF.

## 2.5 Isolation and culture of cells

PC12 cells were cultured in RPMI 1640 (Gibco, USA) with 10% fetal calf serum, 2 mM glutamine, 100 U/mL penicillin (PanEko, Russia), and 100 μg/mL streptomycin (PanEko, Russia). Cells were seeded onto four-well plates and incubated at 37˚C and 5% $CO_2$ for 6 days, after which the medium was replaced with that containing the factors studied. The first plate contained the medium alone and served as the control. In the second and third plates, the medium included 10 ng/mL of Pro-mGDNF and mGDNF, respectively. In all cases, the medium was replaced every 48 h. Recombinant recGDNF (10 ng/mL) (PeproTech, USA) was used as positive control.

E15 rat embryos were used to obtain embryonic spinal ganglia. The embryos were placed into the 100 mm dish with DMEM/F12 (Gibco, USA), 2 mM L-glutamine (PanEko, Russia), 10% fetal calf serum (HyClone, USA), and 50 μg/mL gentamicin (PanEko, Russia). The region of the spinal column with the ganglia was washed twice with Hanks solution (PanEko, Russia) containing 100 μg/mL gentamicin. The spinal ganglia were isolated from their loci under a binocular microscope and placed into a 35 mm dish with the medium drops (100 μl) and incubated at 37˚C and 5% $CO_2$. In another experimental series, the ganglia were divided into 3–5 fragments and each fragment was placed into an individual well. The ganglia and their fragments were initially incubated in the growth medium (DMEM/F12, 10% fetal calf serum, 2 mM L-glutamine, 100 μg/mL gentamicin, 0.8% glucose, and 2 μM HEPES) for 3–4 h at 37˚C and 5% $CO_2$. After their attachment, the growth medium was replaced with 200 μl of the fresh growth medium containing 10 ng/mL of mGDNF or Pro-mGDNF, and incubation continued under 5% $CO_2$ at 37˚C for 10 days; the conditioned and control media were replaced every 2–3 days.

Human neuroblastoma cells SH-SY5Y were cultured in a DMEM/HAM F12 medium (1:1) (Gibco, USA) containing 10% heat-inactivated fetal bovine serum, 1 mM sodium pyruvate, 0.1 mM nonessential amino acids, 1.5 g/l sodium bicarbonate, 100 U/mL penicillin, and 100 μg/mL streptomycin. Cells were maintained in a humid atmosphere with 5% $CO_2$ at 37˚C. All experiments were in triplicate.

PC12 cells were kindly provided by Institute of Cytology Russian Academy of Science.

E15 rat embryos were isolated by us from E15 embryos of Sprague-Dawley rats. All surgery was performed under chloral hydrate anesthesia, and all efforts were made to minimize suffering.

SH-SY5Y was kindly provided by Dr. E. Yu. Rybalkina from Laboratory of genetics of tumor cells, Research Institute of Carcinogenesis, N.N. Blokhin FSBI NMITs of Oncology, Ministry of Health, Russia.

## 2.6 Immunocytochemistry

After 10 days of culture, spinal ganglia were fixed in 4% formaldehyde in PBS at 4˚C for 20 min and washed with PBS for 5 min. For immunohistochemical analysis, the primary rabbit polyclonal antibodies against β3-tubulin (Abcam, UK) were diluted 1:100 with PBS containing 0.3% Triton X-100 (Sigma, USA), and 2% normal donkey serum. The incubation with the primary antibodies was carried out at 4˚C for 12 h, which was followed by washing with PBS. The staining with the secondary antibodies was performed at room temperature for 1 h, then with Cy2-conjugated donkey anti-rabbit antibodies (Jackson Immuno Research, USA), which was followed by washing with PBS. The stained ganglia were embedded in glycerol and analyzed

under an inverted microscope Olympus IX81.222 using blue excitation light and yellow trans-mittance filter (U-MWB). Subsequently, the efficiency of isolated neurotrophic factors was analyzed only for β3-tubulin-positive processes.

### 2.7 Evaluation of neuronal induction by the factors on cell cultures

The activity of the obtained neurotrophic factors was quantified by counting the processes of spinal ganglion neurons stained with antibodies against β3-tubulin. The number of crosses was counted for the processes with diameters of 500 and 750 μm using "Multi-point" tool in ImageJ program.

### 2.8 MTT test

Cells were cultured for 2 days and exposed to MPP+. The effect of the obtained factors (mGDNF or Pro-mGDNF) on the survival rate of immature cells exposed to MPP+ was evaluated using the MTT test, all analyses were carried out in triplicate. In a preliminary experiment, the toxic effect of different MPP+ concentrations (2, 3, and 4 mM) was evaluated. The cell survival rate was about 50% for 2 and 3 mM. Cells were plated into 96-well plates (10,000 cells per well) in 90 μl of the standard culture medium. After 20-h incubation in a humidified atmosphere (at 37°C and 5% $CO_2$), the cultures were supplemented with either MPP+ in the serum-free medium or the medium alone (control) and incubated for 48 h (at 37°C and 5% $CO_2$). Then, 20 μl of MTT (5 μg/mL in saline) was added; 3 h later the solution was discarded and 60 μl of DMSO was added. The amount of formazan was quantified using a multichannel photometer with a 492 nm filter. Cell viability was evaluated by the optical density in the experimental (mGDNF or Pro-mGDNF) and control wells. Cells were cultured in DMEM/F12 with 10% fetal calf serum. The MTT assay analysis was completed using i-control 1.10 software.

### 2.9 Quantitative PCR

After neuroblastoma cells were exposed to the GDNF factors, RNA was isolated using the TRI-zol method (Invitrogen, USA) following the manufacturer's protocol. RNA samples were treated with DNase I (Fermentas, USA) following the manufacturer's protocol. The MMLV reverse transcriptase (Evrogen, Russia) and the oligo(dT)16 primer were used to synthesize cDNA, which was used as the template in quantitative real-time PCR. The SYBR Green I kit (Syntol, Russia) was used for detection. PCR amplification included 45 cycles of denaturation at 94°C for 20 s, annealing at 60°C for 20 s, and elongation at 72°C for 30 s. Each sample was carried out in triplicate. Fluorescence intensity was measured after the elongation stage. Melt curves were plotted after PCR to confirm the homogeneity of amplification products, which were also visualized in 1% agarose gel. The results of quantitative real-time PCR were analyzed using the software supplied with the CFX96 thermocycler (BioRad, USA). Primer sequences were designed using RTPrimerDB and Primer-Blast (NCBI). The efficiency of designed primers was tested by agarose gel analysis of the PCR products. The human housekeeping *HPRT* gene was used as a reference.

The following primers were used:

*B3Tub_F*, 5'-GCGAGATGTACGAAGACGAC-3';

*B3Tub_R*, 5'-TTTAGACACTGCTGGCTTCG-3'.

## 2.10 Generation of model animals and bilateral administration of studied factors

The mouse model of Parkinson's disease was generated as described previously [21]. The work had applied a "chemical model" of C57Bl/6 mouse Parkinson's disease, which was obtained by subcutaneous injection of 40 mg/kg of dopaminergic proneurotoxin 1-methyl-4-phenyl-1,2,3,6-tetrahydropyridine (MPTP) into these mice [38,39]. All *in vivo* experiments were approved by the Ethics Committee of Moscow State University.and were carried out in accordance with "The Guidelines for Manipulations with Experimental Animals" (the decree of the Presidium of the Russian Academy of Sciences of April 02, 1980, no. 12000–496), and the Guidelines for Humane Endpoints for Animals Used in Biomedical Research, Regulations for Laboratory Practice in the Russian Federation. All surgery was performed under chloral hydrate anesthesia, and all efforts were made to minimize suffering. C57BL/6j mice at the age of 2.5–3 months weighing 25–30 g were used in three groups, N = 5–7 per group (the number of animals per group is sufficient to achieve reliable results). Three isoforms of GDNF, namely cmGDNF, mGDNF1-Coli, mGDNF2-Coli, were administered to each group of animals, respectively.

Prior to bilateral administration of recombinant mGDNF, produced by E. coli, animals were anesthetized by chloral hydrate and placed in a stereotaxic frame. Cranial openings 1 mm in diameter were made at coordinates AP 0 mm and mL 2.5 mm bilaterally. The needle was inserted to a depth of 3 mm and 3 µl of isoforms GDNF were injected for 10 min with a gradual needle withdrawal to the depth of 1.5 mm. Thus, isoforms GDNF were injected into the caudate-putamen of mice. Twenty hours later, 40 mg/kg MPTP (Sigma, USA). was administered by subcutaneously injection.

## 2.11 Analysis of MPTP effects in model animals. Motor coordination tests. Immunohistochemical analysis of cranial sections

Three days after the bilateral intrastriatal injection of GDNF, the dopaminergic proneurotoxin MPTP (40 mg/kg) was subcutaneously administered. Control animals were administered with MPTP without a preliminary injection of GDNF (N = 5) (the number of animals per group is sufficient to achieve reliable results). After 2 weeks, the motor coordination of experimental animals was evaluated in the rotarod test (TSE, Germany). Animals were exposed to 6 rpm for 10 min, after which the rotational speed was increased in steps of 1 rpm every 30 s until the animal fell onto the tray with wood shavings. The time of falling and velocity were recorded. The animals were re-anesthetized and perfused through the heart with PBS and then with 4% formaldehyde in PBS. The brain was isolated, fixed again in formaldehyde for 12 h at 4°C, and soaked in 30% sucrose in PBS for 24 h. The cryotome coronal sections of the brain (40 µm) were mounted in PBS. Four series of sections were prepared for each brain sample. The sections in the antifreeze solution were stored at -20°C until staining.

Every fourth section containing the substantia nigra was immunohistochemically stained for tyrosine hydroxylase (TH) using monoclonal antibodies against TH (T2928, Sigma, USA) diluted 1:200 in PBS with 2% normal horse serum, 0.5% Triton X-100 (Sigma, USA), and 0.01% sodium azide (Sigma, USA). Free-floating sections were incubated with agitation in the primary antibodies at 4°C for 48 h. After the reaction with biotinylated horse anti-mouse antibodies (Vector Laboratories, USA) and then in ABC (Vector Laboratories, USA), the standard staining for peroxidase was performed using PBS with 0.03% diaminobenzidine (Sigma, USA) and 0.01% hydrogen peroxide. The stained sections were mounted on slides in 50% glycerol and covered with slips.

TH-immunopositive (TH+) cells were quantified on an Olympus IX81 microscope with a computer-controlled motorized stage (Märzhäuser, Germany) and an Olympus DP72 digital camera. Cells were counted using the Cell* software (Olympus Soft Imaging Solution, Germany). After obtaining an overview of the compact part of the substantia nigra (SNpc) and the ventral tegmental area (VTA) at low magnification (10x objective), TH+ cells were counted using the optical fractionator method at higher magnification (40× objective). The 50×50 μm counting frame was shifted in 200 μm steps in both X- and Y-directions within the ventral part of the midbrain. At each position of the counting frame, the focal plane was shifted in the Z direction by 30 μm. An uninformed operator counted unstained nuclei of TH+ cells in counting frames.

### 2.12 Statistical analysis

Data are presented as mean ± SD. Statistical analysis was performed using the SPSS software. The values were compared by one-way ANOVA followed by Tukey's multiple comparisons test. Statistical significance was accepted at $p < 0.05$.

## 3. Results

The goal of this study was to evaluate the functional significance of mGDNF as a neuronal inducer as well as its dependence on the producer, *E. coli* and mammalian (HEK293) cells. In a previous study [20], we also used Pre-mGdnf/HEK and Pro-mGdnf/HEK; accordingly, we tried to express these recombinant proteins in *E. coli* as well. In the Table 1 the names of the growth factors and the sources of proteins were mentioned.

### Expression of recombinant protein mGDNF/Coli, Pre-mGDNF/Coli, and Pro-mGDNF/Coli

The recombinant mGDNF/Coli, Pre-mGDNF/Coli, and Pro-mGDNF/Coli were isolated from Rossetta (DE3) *E. coli* cells after cloning the mGDNFstp/pET15b, Pre-mGDNFstp/pET15b, and Pro-mGDNFstp/pET15b expression constructs. The techniques were developed to produce active recombinant mGDNF and Pro-mGDNF from *E. coli* (see Materials and methods). The sizes of the obtained proteins correspond to the calculated values: mGDNF, 16kDa; Pre-mGDNF, 18kDa; and Pro-mGDNF, 22kDa. We failed to find conditions for producing active Pre-mGDNF/Coli in *E. coli*; and since this mGDNF form was developed only to analyze the pro-sequence role in GDNF activity, further research was limited to mGDNF и Pro-mGDNF.

**Table 1. The following growth factors were used for the neurite outgrowth assays and analysis of protective properties in mouse MPTP model of PD.**

| № | Growth factor | | Source of protein |
|---|---|---|---|
| (1) | mGDNF/Coli: | | *E. coli*-derived recombinant mGDNF: |
| | | mGDNF1-Coli | recombinant mGDNF isolated from *E. coli* using the standard purification |
| | | mGDNF2-Coli | recombinant mGDNF, isolated from *E. coli* with refolding |
| (2) | Pre-mGDNF/Coli | | *E. coli*-derived recombinant Pre-mGDNF |
| (3) | Pro-mGDNF/Coli | | *E. coli*-derived recombinant Pro-mGDNF |
| (4) | mGDNF/HEK | | the medium conditioned by transgenic mGDNF-producing HEK293 cells containing recombinant mGDNF |
| (5) | Pro-mGDNF/HEK | | the medium conditioned by transgenic mGDNF-producing HEK293 cells containing recombinant Pro-mGDNF |
| (6) | cmGDNF | | recombinant mGDNF, concentrated from the medium conditioned by transfected mammalian cells HEK293 |
| (7) | HEK/GDNF | | HEK293 cells transfected with mGDNF |
| (8) | recGDNF | | recombinant GDNF (PeproTech,USA) |

The results of the induced synthesis of recombinant mGDNF/Coli and Pro-mGDNF/Coli in *E. coli* Rosetta(DE3)/mGDNFstp/pET15b and Rosetta(DE3)/Pro-mGDNFstp/pET15b cells are given in Fig 2a and 2b, respectively.

Fig 2a and 2b also demonstrate the accumulation of insoluble mGDNF/Coli and Pro-mGDNF/Coli proteins (in inclusion bodies), which requires denaturing conditions for their isolation and refolding (renaturing). Additional data are given in S1 Fig.

The recombinant proteins mGDNF/Coli and Pro-mGDNF/Coli were isolated and purified by breaking the cell wall of the producer strain, after which the inclusion bodies were isolated. To solve the identified problem of the recombinant protein aggregation, the refolding of the proteins in buffers with high DTT concentrations and pH. High pH (10) and DTT concentration (100 mM) in the dialysis buffer proved to prevent protein aggregation. Thus, about 2 mg of the purified refolded protein was isolated from 0.5 l of *E. coli* culture.

All methods used to isolate recombinant mGDNF/Coli and Pro-mGDNF/Coli proved inefficient for the production of Pre-mGDNF. The induced induction was insignificant and hardly detectable so that no reproducible experiments could be carried out. Further studies focused on mGDNF/Coli and Pro-mGDNF/Coli.

## Recombinant protein mGDNF/Coli, Pro-mGDNF/Coli and cmGDNF induced neurite outgrowth

PC12 cells were used to analyze the inductive capacity of recombinant mGDNF/Coli and Pro-mGDNF/Coli produced in *E. coli*. PC12 cell line originates from a pheochromocytoma of the

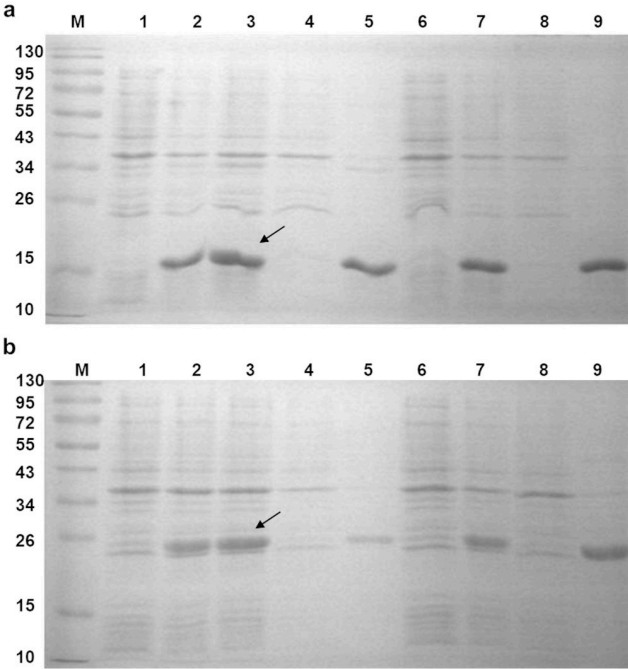

**Fig 2.** (a) Recombinant mGDNF synthesis and localization in *E. coli* Rosetta(DE3)/*mGDNF*stp/pET15b cells. (b) Recombinant Pro-mGDNF synthesis and localization in *E. coli* Rosetta(DE3)/*Pro-mGDNF*stp/pET15b cells. M, calibration standard; 1, prior to induction at 25˚C; 2, induction after 17-h incubation at 25˚C; 3, induction after 24-h incubation at 25˚C; 4, supernatant after incubation at 25˚C; 5, inclusion bodies after incubation at 25˚C; 6, prior to induction at 37˚C; 7, induction after 3-h incubation at 37˚C; 8, supernatant after incubation at 37˚C; 9, inclusion bodies after incubation at 37˚C. Arrows indicate expressed proteins. The synthesis of recombinant proteins by producer strains was induced by 1 mM isopropyl-β-d-thiogalactoside and cells were cultured under the selected optimal conditions: Rosetta(DE3)/mGDNFstp/pET15b, 3.5 h at 37˚C and 250 rpm; and Rosetta(DE3)/Pro-mGDNFstp/pET15b, 24 h at 25˚C and 250 rpm.

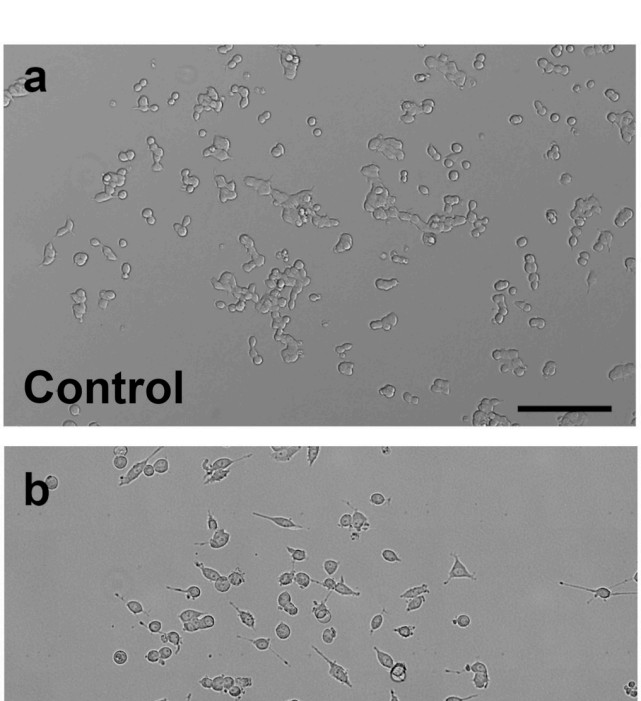

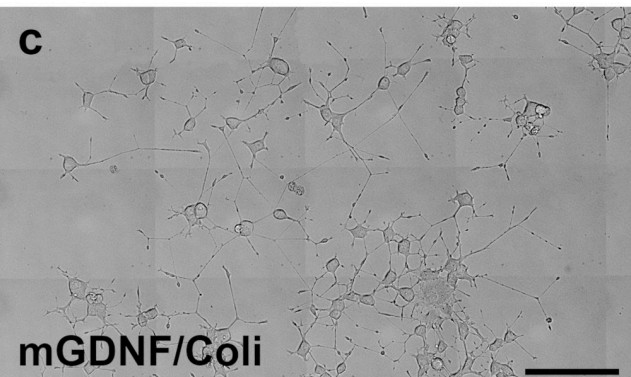

**Fig 3. Effect of mGDNF/Coli and Pro-mGDNF/Coli on PC12 cells.** After 6 days of PC12 culture, the medium was supplemented with 10 ng/mL of the studied factors, mGDNF/Coli or Pro-mGDNF/Coli. (**a**) control (PC12 culture without the factors); (**b**) PC12 culture with Pro-mGDNF/Coli; (**c**) PC12 culture with mGDNF/Coli. Scale bar 500 μm.

rat adrenal medulla, which is derived from embryonic neural crest composed of neuroblasts and eosinophilic cells. PC12 is a universal cell system in pharmaceutical studies of drug efficiency and is easy to culture.

The proportion of cells with neural processes in cultures exposed to the factors was significantly greater than the control. No processes were formed in control cells (Fig 3a), while short processes could be observed 4 days after *E. coli*-derived Pro-mGDNF/Coli was added to the medium, which indicates a minor neural inductive capacity of the factor (Fig 3b). This observation significantly differs from the activity of Pro-mGDNF/HEK synthesized in transgenic

HEK293 [20] that blocked the process formation in PC12 cells. Thus, the properties of Pro-mGDNF synthesized in *E. coli* and HEK293 cells substantially differ.

Four days after mGDNF/Coli was added to the medium, long branched processes could be observed, which indicates a significant neural inductive capacity of mGDNF/Coli (Fig 3c).

Rat embryonic spinal ganglion cells [20] were used to study the inductive capacity of the factors on neuronal cells.

To verify the neural inductive capacity of mGDNF/HEK described previously, we used cmGDNF isolated from the medium condition by HEK293 cells using affinity purification using GFP-tagged proteins and GFP-TRAP agarose. Thus, in contrast to the previous paper [20], we used the immunoprecipitation-isolated protein rather than a conditioned medium. Four days after the addition of cmGDNF induced the formation of numerous long processes, which demonstrates the high efficiency of this protein as a neural inducer (Fig 4a). Rat

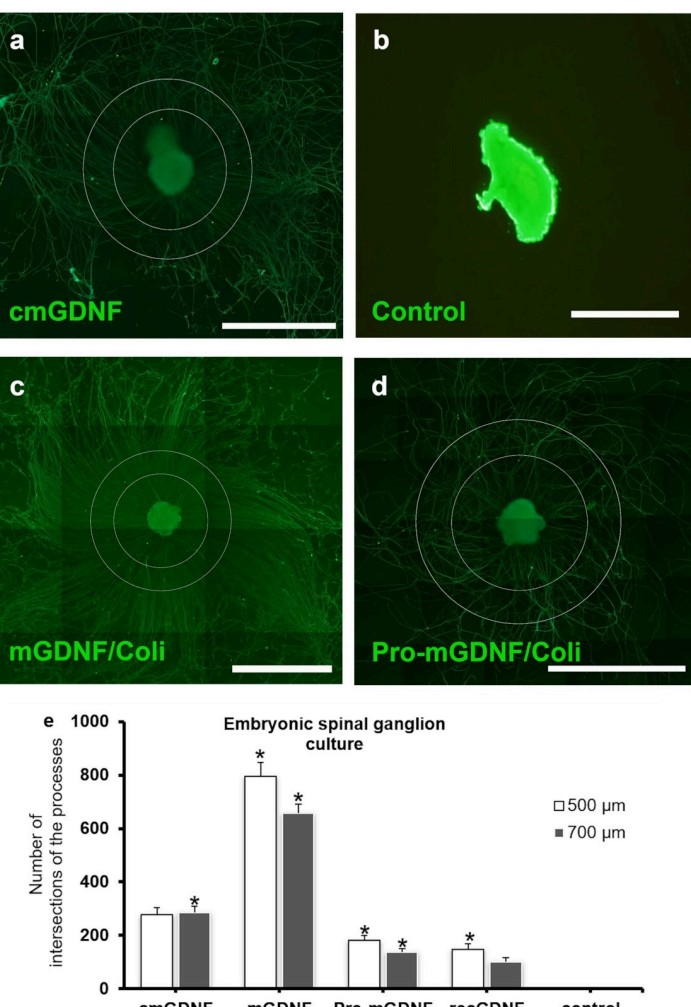

**Fig 4.** (a), (b), (c) and (d) Analysis of the isolated factors using the rat model of whole embryonic spinal ganglion culture supplemented with: (a) cmGDNF (mGDNF concentrated from the medium conditioned by transgenic cells); (b) no factors (control); (c) recombinant mGDNF/Coli; (d) recombinant Pro-mGDNF/Coli. Scale bar 500 μm. (e) Quantitation of ganglion cell processes. cmGDNF was isolated by immunoprecipitation from the medium conditioned by HEK293 cells; mGDNF, recombinant mGDNF/Coli; Pro-mGDNF, recombinant Pro-mGDNF/Coli; recGDNF, positive control; control, negative control with no factors. The results are presented as the mean ±SD (*p<0,05 vs control) (N = 3).

embryonic spinal ganglion cells cultured under the same conditions but without the GDNF variants demonstrated no process formation (Fig 4b). The addition of the recombinant mGDNF/Coli from *E. coli* also induced the formation of long branched processes (Fig 4c).

The results of the exposure of spinal ganglion culture to Pro-mGDNF/Coli were equally unforeseen as of PC12 cells. Pro-mGDNF/HEK synthesized by mammalian cells demonstrated no new inductive properties [20], while this factor isolated from *E. coli* here unexpectedly demonstrated the capacity to induce process formation using the whole spinal ganglion model. Pro-mGDNF/Coli induced the formation of shorter and less voluminous processes compared to both positive cmGDNF control and *E. coli*-derived mGDNF/Coli; however, the process formation induced by Pro-mGDNF/Coli was more pronounced (Fig 4d).

The comparative efficiency of the obtained variants of neurotrophic factors was quantified by counting processes of spinal ganglion cells. Crossings between processes 500 and 750 μm in diameter were counted on the micrographs (Fig 4a–4d). Fig 4e visualizes this analysis. The highest capacity to induce process formation in the embryonic spinal ganglion was observed for the recombinant mGDNF/Coli synthesized in *E. coli*. CmGDNF concentrated from the conditioned medium also demonstrated a positive effect. The next efficient factor was Pro-mGDNF/Coli; and recGDNF was the least efficient. It should be noted that the exposure to cmGDNF from HEK293 cells induced a smaller number of processes but they were longer; and only cmGDNF induced the formation of longer processes. After adding cmGDNF, there is an equal number of intersections on the small and large diameter counting circles. On the contrary, adding other forms of GDNF have yielded more intersections on the small circle relative to the large one. Perhaps due to activity of factors, the fibers begin to branch earlier and do not grow to a large circle.

Notice also that recGDNF demonstrated a much lower efficiency as a neuronal inducer similarly to that of Pro-mGDNF/Coli.

Analysis of the impact of the recombinant factors in the embryonic dissociated ganglion model yielded results similar to the previous model. In the absence of GDNF, rat dissociated embryonic spinal ganglion cells formed no processes after culturing for 10 days (Fig 5a). At the same time, the isolated factors mGDNF/Coli (Fig 5b) and Pro-mGDNF/Coli (Fig 5c) induced the process formation. As with the whole ganglion, the exposure to Pro-mGDNF/Coli from *E. coli* was less efficient than mGDNF/Coli in the process formation.

The above results were obtained on rat cells and ganglia. SH-SY5Y human neuroblastoma cells were used to study the effect of the obtained factors on the survival of human neural cells. The culture media were supplemented with 1-methyl-4-phenylpyridinium ion (MPP+), which is used to simulate neurodegenerative processes such as Parkinson's disease (Fig 6a) [40,41].

This model was used to evaluate the impact of the factors on cell survival using three concentrations of *E. coli*-derived mGDNF/Coli and Pro-mGDNF/Coli: 10, 20, and 50 ng/mL. At 2 mM MPP, the effects of mGDNF/Coli or Pro-mGDNF/Coli were insignificant (Fig 6c). Conceivably, Pro-mGDNF/Coli was a more efficient factor of neuroblastoma cell survival. As the concentration of MPP increased to 3 mM (Fig 6d), the effect of mGDNF/Coli and Pro-mGDNF/Coli on cell survival was equalized and varied within 52–54% as against 47% in control, which can be considered as an insignificant positive effect.

The conditions were aggravated by increasing the exposure to MPP+ to 48 h. The obtained results demonstrated no improvement in the viability of cells treated with mGDNF/Coli and Pro-mGDNF/Coli, while the recombinant recGDNF increased the survival rate by 5–13% (Fig 6e).

Thus, it is fair to say that these two factors synthesized in *E. coli* do not maintain the viability of human neuroblastoma SH-SY5Y cells after toxic exposure. The same cell line was used to analyze the neural inductive capacity of mGDNF/Coli and Pro-mGDNF/Coli from *E. coli*.

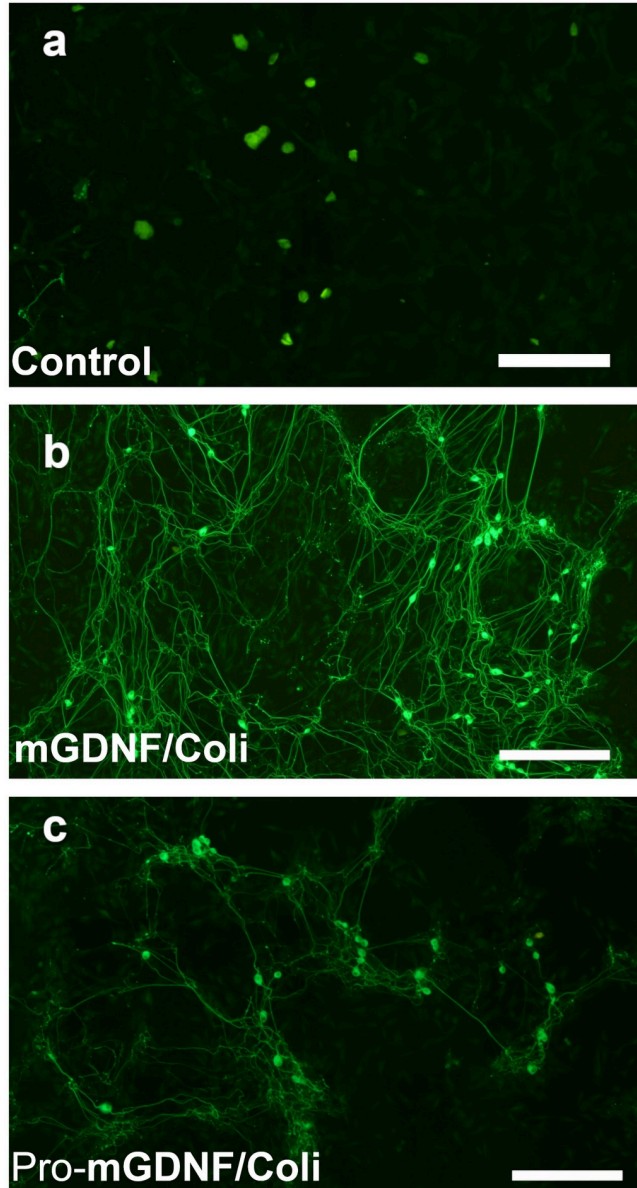

**Fig 5. Effect of recombinant GDNF factors on the rat dissociated embryonic spinal ganglion.** All dissociated embryonic ganglia were immunohistochemically stained for β3-tubulin to confirm the neuronal origin of the processes; and all processes were β3-tubulin-positive. (**a**) control, no GDNF; (**b**) mGDNF/Coli; (**c**) Pro-mGDNF/Coli. Scale, 500 μm.

Similarly, neuroblastoma cells were exposed to the obtained factors without MPP+ pretreatment. This experiment was made to test the capacity of these factors to induce neural differentiation. Real-time quantitative polymerase chain reaction (RT-qPCR) was applied to evaluate the changes in the neural marker expression using primers for β3-tubulin.

This experiment demonstrated a significant increase in β3-tubulin expression in cells exposed to the recombinant mGDNF/Coli (Fig 6b).

The exposure to Pro-mGDNF/Coli also insignificantly increased the marker expression relative to control.

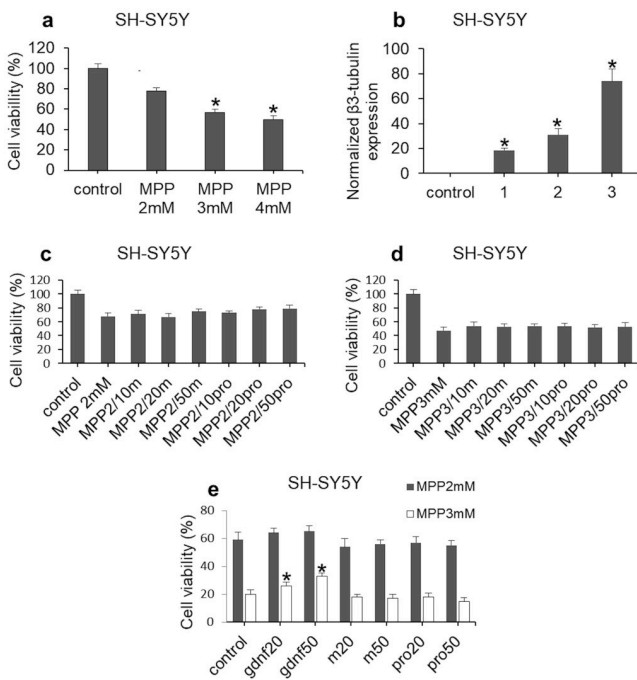

**Fig 6.** (a) Preliminary evaluation of MPP+ toxic effect on human neuroblastoma SH-SY5Y cell survival. The dose-dependence and high reproducibility were demonstrated. (b) Analysis of β3-tubulin expression in cultured SH-SY5Y neuroblastoma cells exposed to the factors: 1, recGDNF; 2, Pro-mGDNF/Coli; 3, mGDNF/Coli. (c) and (d) Effect of the obtained recombinant GDNF factors on SH-SY5Y neuroblastoma cells after the exposure to MPP+ for 30 h. SH-SY5Y was incubated with different concentrations of GDNF factors (10, 20, 50 ng/mL), (c) 2 mM MPP+; (d) 3 mM MPP+. (e) Effect of recGDNF, mGDNF/Coli, and Pro-mGDNF/Coli on the viability of SH-SY5Y neuroblastoma cells at 48-h exposure to MPP+; gdnf-, recombinant factor recGDNF derived from the full-length GDNF; m, mGDNF/Coli; pro, Pro-mGDNF/Coli. Data represents the mean ± SEM (for (a), (b) and (e) *p<0,05 vs. control) (N = 3).

## Analysis of protective properties in mouse MPTP model of PD

The model of chemically induced death of dopaminergic neurons (as in Parkinson's disease) by subcutaneous MPTP injection was used to analyze the protective properties of *E. coli*-derived mGDNF/Coli *in vivo* (N = 5–7 per group). TH-immunopositive cells were counted on the midbrain sections prepared 14 days after the injection of 40 mg/kg MPTP into animals bilaterally intrastriatally pre-administered with the factors (Fig 7).

Several mGDNF variants were tested in the experiment:

1. cmGDNF concentrated from the medium conditioned by transfected mammalian cells HEK293;

2. mGDNF/Coli isolated from *E. coli* using the standard purification (mGDNF1-Coli);

3. mGDNF/Coli, isolated from *E. coli* with refolding (mGDNF2-Coli).

The effects of administration of these factors prior to MPTP exposure were analyzed vs intact animals, control MPTP-injected animals without any factors, and control animals injected with transgenic mGDNF/GFP or GFP cells prior to MPTP exposure [21].

Among the three studied factors, the highest neuroprotective capacity was observed in cmGDNF isolated from the medium conditioned by transgenic mammalian cells (HEK293/mGDNF/GFP) (Fig 8a), although it was somewhat lower compared to bilaterally injected transgenic HEK293/mGDNF/GFP cells [21]. No neuroprotective activity was observed after

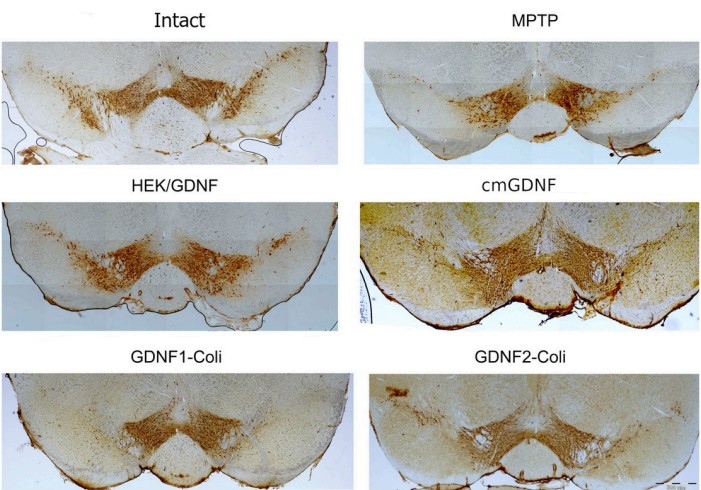

**Fig 7. Immunohistochemical staining for tyrosine hydroxylase of the ventral midbrain in intact animals and those 14 days after the exposure to MPTP pre-administered with the recombinant mGDNF/Coli or cmGDNF 20 h prior to MPTP exposure or HEK/GDNF injection into the striatum 3 days prior to MPTP exposure.** HEK/GDNF, HEK293 cells transfected with mGDNF; cmGDNF, mGDNF concentrated from medium containing mGDNF; GDNF1-Coli, mGDNF isolated from *E. coli* by the standard method; GDNF2-Coli, mGDNF isolated from *E. coli* with refolding using the mouse serum.

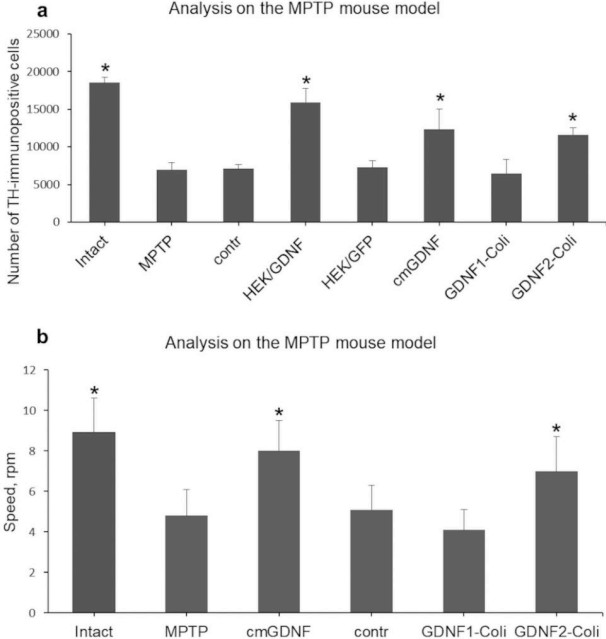

**Fig 8.** (a) The number of tyrosine hydroxylase-immunopositive neurons in the ventral midbrain in intact animals and those pre-administered with the GDNF variants into the striatum 20 h prior to MPTP exposure for 14 days. (b) Motor coordination of experimental animals administered with recombinant mGDNF and control saline solution with the subsequent MPTP injection. The Contr and GDNF1-Coli values significantly differ (p≤0,05) from those of cmGDNF and и GDNF2-Coli. intact, no exposure; MPTP, exposure to MPTP alone; Contr, control saline solution without GDNF; HEK/GDNF, HEK293 cells transfected with mGDNF; HEK/GFP medium conditioned by GFP-transgenic cells; cmGDNF, mGDNF concentrated from the conditioned medium; GDNF1-Coli, mGDNF isolated from *E. coli* by the standard method; GDNF2-Coli, mGDNF isolated from *E. coli* with refolding using the mouse serum. The results are presented as the mean ±SD (*p<0,05 vs control) (N = 5–7 per group).

the administration of mGDNF/Coli isolated from *E. coli* by the standard method (mGDNF1-Coli). The number of TH-immunopositive cells after mGDNF1-Coli administration in the MPTP experiment was lower even compared to the control with no mGDNF1-Coli administration. At the same time, the number of TH-immunopositive cells was significantly (twice) higher after mGDNF2-Coli administration compared to control and mGDNF1-Coli. Thus, mGDNF/Coli isolated from *E. coli* with refolding using the recipient animal serum (mGDNF2-Coli) demonstrated a significant neuroprotective effect.

The analysis of the motor activity of control and experimental mice demonstrated better motor coordination in animals injected with cmGDNF concentrated from the conditioned medium or recombinant mGDNF/Coli (GDNF1-Coli) isolated from *E. coli* with refolding using the mouse serum (Fig 8b).

## 4. Discussion

Initially, GDNF was considered a promising agent to treat diseases associated with the death of central and peripheral neurons. GDNF was also regarded as the main remedy for neurodegenerative disorders, particularly Parkinson's disease, as well as other resources causing neuronal death such as ischemic stroke or brain trauma. In addition, several GDNF modifications were found [42] and an isoform without to the properties required for neuronal regeneration could be used in clinical trials [20]. It should be noted that, apart from the identification of GDNF isoforms optimal for neuronal regeneration, it is critical to compare the function of GDNF synthesized in *E. coli* and mammalian cells. Interestingly, two studies published contiguously [20,43] demonstrated a high neural inductive capacity of GDNF lacking the N-terminal sequence. In this context, we focused on the neural inductive capacity of N-truncated GDNF as well as on the functional difference between the N-truncated proteins synthesized in *E. coli* and mammalian cells, specifically, HEK293 cells. On the one hand, the *E. coli* production system is very attractive considering its high productivity and low cost. On the other hand, this system is incapable of certain post-translational modifications (such as glycosylation), proteolytic maturation, and disulfide bond formation in the synthesized protein. Thus, certain medically relevant proteins cannot be produced in *E. coli* due to improper folding or glycosylation. The problem of the inexpedient application of *E. coli* for the production of recombinant proteins has been discussed for some time in various publications. It has already been shown that physiological properties of *E. coli* limit their use for the production of proteins in their native form, especially polypeptides that are subjected to major post-translational modifications [44]. Recent articles assess the change in the properties of specific proteins produced by different producer cells. For example, in the work of Yu et al [45], such an analysis carried out for Recombinant human IFNα2b (rhIFNα2b) synthesized by various producers demonstrated the inexpediency of using yeast cells.

The initial assumption was that *E. coli*-derived GDNF is functional and thus produced recombinant protein was used for a long time. However, unsuccessful clinical trials question this assumption. As a consequence, communications appeared suggesting to utilize GDNF produced in mammalian cell systems to provide natural glycosylation and phosphorylation. Here, we used the truncated GDNF form described previously, mGDNF [20]. We compared the neuroinductive properties of recombinant *E. coli*-derived mGDNF and HEK293-derived cmGDNF as well as of the medium conditioned by transgenic mGDNF-producing HEK293 cells. Considering the observed low efficiency of *E. coli*-derived mGDNF compared to HEK293-derived one, we developed new mGDNF folding conditions with the protein dialyzed in the presence of the animal recipient serum, which significantly improved the neuroinductive properties of the isolated protein. Our results are briefly summarized in Table 2.

**Table 2. The analysis of biological effects of GDNF produced by bacterial expression system (*E. coli*) and mammalian expression system (HEK293) [20,21].**

| Protein modification | Neural positive processes in PC12 cells [20,21] | Neural positive processes in embryonic spinal ganglion cells [20] | Neural positive processes in dissociated embryonic spinal ganglion cells [20] | TH-positive cells *in vivo* [21] |
|---|---|---|---|---|
| Medium conditioned by HEK293 cells expressing mGDNF | ++ | +++ | +++ | +++ |
| Medium conditioned by HEK293 cells expressing Pro -mGDNF | ½+ | - | - | n/a |
| Medium conditioned by HEK293 cells expressing Pre-mGDNF | ½+ | ++ | + | n/a |
| recombinant mGDNF (*E. coli*) | + | ++ | ++ | + (new folding conditions) |
| recombinant Pro-mGDNF (*E. coli*) | + | + | ½+ | n/a |
| cmGDNF (HEK293) | n/a | +++ *long branched processes relative to recombinant mGDNF | +++ | ++ |
| negative control | - | - | - | - |
| positive control, commercial recGDNF | + | ½+ | ½+ | + |

+ detected biological activity in reference to negative control.

n/a no data.

We have found that the neuroinductive properties of mGDNF and Pro-mGDNF notably vary with the producer, HEK293 or *E. coli*. Previously, Kust et al. [20] demonstrated a high neural inductive capacity of mGDNF/HEK produced in HEK293 in an *in vitro* model with embryonic spinal ganglion, while media conditioned by Pro-mGDNF-expressing cells not only missed such capacity but largely blocked the formation of β3-tubulin-positive processes in embryonic spinal ganglia. At the same time, mGDNF/Coli production in *E. coli* modified its properties: it induced the formation of neural processes but their branching substantially differed from that induced by HEK293-derived mGDNF/HEK. To our surprise, *E. coli*-derived Pro-mGDNF/Coli had altered properties as well. It demonstrated low neural inductive capacity unlike HEK293-derived Pro-mGDNF/HEK lacking this capacity [20]. The fact that only cmGDNF induced the formation of longer processes compared to other studied GDNF isoforms deserves special attention and suggests the possibility of using the isoforms of GDNF in different clinical cases/nosology. After peripheral nerve injuries, it is important that the regenerating axons reach their target as quickly as possible [46]. In this regard, the effect of cmGDNF may represent a promising tool for the damaged nerves recovery therapy [46,47]. In case of a violation of local connections between neurons, for example, during ischemia, secondary progressive multiple sclerosis, or neurodegenerative diseases, rapid recovery of the network is required [48–50], which can be provided by stimulating the formation of the neuronal short and branched processes.

Moreover, previously we have found that Pro-mGDNF/HEK almost completely blocked neuronal process formation in rat embryonic spinal ganglion, which suggests the need of a more thorough analysis of GDNF isoforms as neuroinducers, since a wrong choice of an isoform can lead to a neural differentiation blockade. Overall, this questions using *E. coli* cells as the GDNF producer since the synthesized products have altered properties and can be at least useless in such therapy and at most can have unexpected and/or harmful consequences.

The study of the ability of the obtained factors to maintain the viability of cells *in vitro* on the human neural cell culture SH-SY5Y demonstrated low activity of *mGDNF* and *Pro-mGDNF*.

We can even state that Pro-mGDNF (GDNF with the deleted Pre-region) showed more significant results as a factor supporting the viability of SH-SY5Y neuroblastoma cells.

Besides, the investigation of the neuroinductive properties of recombinant mGDNF and Pro-mGDNF on the SH-SY5Y neuroblastoma cell culture showed an insignificant efficacy of Pro-mGDNF and a more significant effect of mGDNF. The results demonstrate that mGDNF isolated from *E. coli* loses the ability to maintain cell viability, but has good neuroinductive properties *in vitro*.

In addition, the analysis of mGDNF and Pro-mGDNF compared to the recombinant recGDNF protein showed that the results obtained for factors with a deleted pro-region are not characteristic for "normal" GDNF with the pro-region of normal length. It should be noted that this protein is secreted through the Golgi apparatus and may have properties other than the factors obtained in the current study.

Thus, the results obtained confirm our hypothesis that the GDNFs secreted in the body in different ways (through the Golgi apparatus or directly through the vesicles) possess different properties. Probably, GDNF with a long pro-region is required to maintain cell viability, while GDNF with a shortened pro-region (or, in our case, with a deleted one) may stimulate neural differentiation of immature cells. Thus, mGDNF can be applied for neural cells recovery after their traumatic death or in case of neurodegenerative diseases.

Of particular interest is that in the chemical model of Parkinson's disease, the *E. coli*-derived mGDNF/Coli demonstrated a positive but much smaller effect on the ganglia compared to that of HEK293-derived cmGDNF.

Firstly, this indicates the imperfection of the *in vitro* models for the analysis of neuroinductive properties of different molecules. However, this may seem acceptable, because the evaluation of the neuroinductive properties in brain tissue where cells have a different density of interaction with each other is of great importance. Despite the fact that the *in vitro* model is still used due to its simplicity, the conclusions about the effectiveness of a neuroinductive molecule can only be drawn after the consistent *in vitro* and *in vivo* studies. Secondly, the results of the current work once again confirm the importance of using mammalian cells as GDNF producers.

We would also like to note our undoubted success in obtaining a more effective GDNF neuroinducer molecule from *E. coli* using the recipient animal serum. This may indicate the absence of the formation of the necessary bonds (for example, disulfide bonds) in the tertiary structure of the protein in the case of standard refolding.

Thus, the efficiency of a neural inducer should be evaluated in both *in vitro* and *in vivo* models. In addition, this study emphasizes the benefits of cmGDNF produced in mammalian cells more closely replicating the natural conditions.

Further researches focused on the production of modified GDNF isoforms from mammalian cells and investigation their secretion pathways, as well as their interaction with cell receptors are required. The search for small molecules based on GDNF and able pass the blood-brain barrier and analysis of their neurotrophic potential for clinical use in neurodegenerative diseases are to be continued.

## Supporting information

**S1 Fig. (a), (b) Electrophoretic analysis of Pro-mGDNF isolation samples.**
(PDF)

**S1 Raw images.**
(PDF)

## Author Contributions

**Conceptualization:** Galina V. Pavlova.

**Formal analysis:** Dzhirgala V. Shamadykova, Dmitry Y. Panteleev, Nadezhda N. Kust, Ekaterina A. Savchenko, Ekaterina Y. Rybalkina, Alexander V. Revishchin.

**Funding acquisition:** Galina V. Pavlova.

**Investigation:** Dzhirgala V. Shamadykova, Dmitry Y. Panteleev, Nadezhda N. Kust, Ekaterina A. Savchenko, Ekaterina Y. Rybalkina, Alexander V. Revishchin.

**Methodology:** Dmitry Y. Panteleev.

**Resources:** Dzhirgala V. Shamadykova.

**Validation:** Dzhirgala V. Shamadykova, Dmitry Y. Panteleev, Nadezhda N. Kust, Ekaterina A. Savchenko, Ekaterina Y. Rybalkina, Alexander V. Revishchin.

**Writing – original draft:** Dzhirgala V. Shamadykova.

**Writing – review & editing:** Dmitry Y. Panteleev, Nadezhda N. Kust, Ekaterina A. Savchenko, Ekaterina Y. Rybalkina, Galina V. Pavlova.

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
