## [Decision Letter · Decision Letter 0]

26 Feb 2021

PONE-D-20-31228

Neuroinductive Properties of mGDNF  Depend on the Producer, E. Coli or Human Cells

PLOS ONE

Dear Dr. Shamadykova,

Thank you for submitting your manuscript to PLOS ONE. After careful consideration, we feel that it has merit but does not fully meet PLOS ONE’s publication criteria as it currently stands. Therefore, we invite you to submit a revised version of the manuscript that addresses the points raised during the review process.

Please see my comments below.

We look forward to receiving your revised manuscript.

Kind regards,

Sujeong Jang

Academic Editor

PLOS ONE

Journal Requirements:

2. To comply with PLOS ONE submissions requirements, please provide methods of sacrifice in the Methods section of your manuscript.

4.We note that the grant information you provided in the ‘Funding Information’ and ‘Financial Disclosure’ sections do not match.

5.Thank you for stating the following in the Financial Disclosure section:

"G.P. received grant funding. This research was funded by the Ministry of Science and Higher Education of the Russian Federation (grant number 2020-1902-01-327) and RFBR (grant number 18-29-01-012). The funders had no role in study design, data collection and analysis, decision to publish, or preparation of the manuscript. "

We note that one or more of the authors are employed by a commercial company: Ltd Apto-pharm

Additional Editor Comments:

This paper described that prokaryotic and eukaryotic has a potential to rescue a disease to analyze the significance of mGDNF synthesis in mammalian or E. coli cells. It is very interesting research and quite novel findings. However, there are several revision for submission.

1. In Materials and Methods, you have to explain or describe the reference why you studied and did the experiment in this study? If you have references, you have to mention that for understanding.

2. In Materials and Methods 2.3, you have to describe the cells what you used in here.

3. Please provide information on the sample sizes used, including a justification for this number. Where relevant, report the number of independent replications for each experiment. For more details please see here: https://journals.plos.org/plosone/s/submission-guidelines.#loc-statistical-reporting

4. You have to write ‘mL’ instead of ‘ml’, for example, ‘The culture broth of the producers of recombinant mGDNF (750 ml) was cooled…….’ -> ‘The culture broth of the producers of recombinant mGDNF (750 mL) was cooled…..’.

5. In Materials and Methods 2.5, you used PC12, E5 rat embryos, SH-SY5Y cells in here, but you did not mentioned the source of cells. Where did you get the cells? If you buy it, you have to mention it. If you get them from other researcher, you have to describe it.

6. In Figure 5, you have to stain nuclei with DAPI or others for distinguish.

7. In discussion, I want to know what the next is. You have to suggest of the next step to use your technique or the data.

Reviewers' comments:

Reviewer's Responses to Questions

**Comments to the Author**

1. Is the manuscript technically sound, and do the data support the conclusions?

Reviewer #1: Partly

Reviewer #2: Partly

2. Has the statistical analysis been performed appropriately and rigorously? 

Reviewer #1: Yes

Reviewer #2: Yes

3. Have the authors made all data underlying the findings in their manuscript fully available?

Reviewer #1: Yes

Reviewer #2: No

4. Is the manuscript presented in an intelligible fashion and written in standard English?

Reviewer #1: Yes

Reviewer #2: Yes

5. Review Comments to the Author

Reviewer #1: The purpose of this manuscript is to demonstrate the feasibility and usefulness in using molecular biology transfection approach to generate mGDNF protein from HEK293 cells. Despite the demonstration, there is no scientific advance in the approach. The outcome is predictable. For this reason, there is no need to publish such a study.

Reviewer #2: In this research article, the authors describe the effectiveness of various forms of glial-derived neurotrophic factor (GDNF) as a neural protective and regenerative agent. In particular, the manuscript describes differences in activity between GDNF produced by and isolated from E. Coli and mammalian (HEK293) cells, across several in vitro assays and one animal model of Parkinson’s Disease. Overall, their findings suggest that the use of GDNF isoforms expressed by HEK293 cells leads to increased neural morphology in vitro. Furthermore, mice treated with mammalian-expressed GDNF isoforms had increased tyrosine hydroxylase activity (implying increased DOPA production) in the brain, and better motor control compared to E. Coli-expressed GDNF isoforms. Moreover, when additional protein folding steps were incorporated for E.Coli-expressed GDNF isoforms, neuroregenerative capacity more closely resembled that of the mammalian-expressed isoforms. Taken together, these results help shed light on previously unexplained inconsistencies in the effectiveness of GDNF in vitro and in clinical trials, and suggests that with careful engineering, production of therapeutic GDNF using E.coli or other prokaryotic systems may be possible.

Overall, the manuscript is well-written, compelling, and the results conveyed seem to constitute a significant contribution to the field. There are a few areas where additional clarity would help the reader better understand the context and impact of the work, as outlined below.

• In the Introduction, the authors state that “Potential causes [of controversial results in GDNF clinical trials] were discussed by Tenenbaum and Humbert-Claude.” However, these potential causes are not summarized until the Discussion. Moving at least some parts of the Discussion, particularly where the potential causes of controversial clinical trial results are described, would help the reader place the work in context from the beginning and help lay a strong foundational rationale for why the work was performed.

• The authors should more thoroughly describe methods used to count the number of intersections between ganglion neuron processes (Section 2.7), including whether the process was automated.

• In Figure 4e, it seems as though there are a greater number of intersections in the smaller diameter counting circle than in the larger one. The authors should explain why this might be the case. (This discrepancy may also be explained by more clear methods in Section 2.7, as suggested in the previous point).

• Commercially-sourced GDNF (“recGDNF”) was used as a control in several experiments. Based on the information provided (e.g., source: Peprotech), this GDNF also appears to have been produced by E.Coli. The authors should describe, to the extent possible (understanding that some of Peprotech’s information may be proprietary), how this protein differs from those produced in-house for the study.

• In reference to Figure 4, the authors state that “…exposure to cmGDNF from HEK293 cells induced a smaller number of processes but they were longer; and only cmGDNF induced the formation of longer processes.” The discussion should include some interpretation of these results, including whether this property is more or less beneficial than numbers of intersections, and in what clinical scenarios length vs. branching, or intersections, may be most relevant.

• Keeping the order of experimental groups consistent between all figures in the manuscript, and/or adding in-figure text labels for each panel, would be helpful for the reader.

• The authors state that all data are contained within the manuscript and supporting information, but individual data points and/or additional images are not included. These should be made available via a public repository or similar, according to PLOS One guidelines.

• In reference to Figure 4, the authors state that “The results…were equally as amazing…” This should be rephrased so as to be objective and technical (syntax).

6. PLOS authors have the option to publish the peer review history of their article (what does this mean?). If published, this will include your full peer review and any attached files.

Reviewer #1: No

Reviewer #2: No

---

## [Author Response · Author response to Decision Letter 0]

25 Apr 2021

Journal Requirements:

We have revised our manuscript in accordance with PLOS ONE's style requirements.

2. To comply with PLOS ONE submissions requirements, please provide methods of sacrifice in the Methods section of your manuscript.

We have added a sentence to section 2.10 (line 266): “All surgery was performed under chloral hydrate anesthesia, and all efforts were made to minimize suffering.” 

We have added to paragraph 2.11 a description of the procedure for killing experimental animals: “The animals were re-anesthetized and perfused through the heart with PBS and then with 4% formaldehyde in PBS.”

We have uploaded the S1_raw_images file containing the original uncropped and unadjusted images underlying all blot or gel data reported in our manuscript. The article presented uncircumcised blot or gel results. All original image data are also available from the Harvard Dataverse Network database (accession number https://doi.org/10.7910/DVN/DIYBUS) (private url https://dataverse.harvard.edu/privateurl.xhtml?token=a01509d7-a3a6-41c5-a646-de9ab3a13dbc)

We made corrections to the ‘Funding Information’ section in accordance with the ‘Financial Disclosure’ section and amended the grant number: instead of grant number 2020-1902-01-327, we have cited “grant number 075-15-2020-809 (13.1902.21.0030)”

“G.P. received grant funding. This research was funded by the Ministry of Science and Higher Education of the Russian Federation (grant number 075-15-2020-809 (13.1902.21.0030) and RFBR (grant number 18-29-01-012). The funders had no role in study design, data collection and analysis, decision to publish, or preparation of the manuscript.”

5.Thank you for stating the following in the Financial Disclosure section:

"G.P. received grant funding. This research was funded by the Ministry of Science and Higher Education of the Russian Federation (grant number 2020-1902-01-327) and RFBR (grant number 18-29-01-012). The funders had no role in study design, data collection and analysis, decision to publish, or preparation of the manuscript. "

We note that one or more of the authors are employed by a commercial company: Ltd Apto-pharm

Please know it is PLOS ONE policy for corresponding authors to declare, on behalf of all authors, all potential competing interests for the purposes of transparency. PLOS defines a competing interest as anything that interferes with, or could reasonably be perceived as interfering with, the full and objective presentation, peer review, editorial decision-making, or publication of research or non-research articles submitted to one of the journals. Competing interests can be financial or non-financial, professional, or personal. Competing interests can arise in relationship to an organization or another person. Please follow this link to our website for more details on competing interests: http://journals.plos.org/plosone/s/competing-interests.

Ekaterina A. Savchenko performed this research in Burdenko NMCNS. Ltd Apto-Pharm did not have any additional role in the study design, data collection and analysis, decision to publish, or preparation of the manuscript, there is no conflict of interest. At the end of 2020, Ltd Apto-Pharm was closed, therefore, we have to remove it from the list of affiliations being proved by Ltd Apto-Pharm representative. Currently Ekaterina A. Savchenko is employee of Burdenko NMCNS, Moscow, Russian Federation.

At the same time DS, DP, AR, GP have moved into Institute of Higher Nervous Activity and Neurophysiology, Russian Academy of Sciences, Moscow, Russian Federation. Therefore, the current affiliation had been added.

We have added a Supporting Information and caption section at the end of the manuscript: “S1 Fig. (a), (b) Electrophoretic analysis of Pro-mGDNF isolation samples.”

We have updated the in-text citation in the section Expression of recombinant protein mGDNF/Coli, Pre-mGDNF/Coli, and Pro-mGDNF/Coli in the Results on line 345: “Additional data are given in S1 Fig.”

Additional Editor Comments:

1. In Materials and Methods, you have to explain or describe the reference why you studied and did the experiment in this study? If you have references, you have to mention that for understanding.

In Methods section , a reference is added to our paper, which describes the technique of subcutaneous injection of MPTP, which causes symptoms of Parkinson's disease in mice. (Revishchin A, Moiseenko L, Kust N, et al (2016) Effects of striatal transplantation of cells transfected with GDNF gene without pre- and pro-regions in mouse model of Parkinson's disease BMC Neurosci 17: https: // doi. org / 10.1186 / s12868-016-0271-x). But considering the requirements of the reviewer, we have included a short explanation in the Section 2.10.

“The mouse model of Parkinson’s disease was generated as described previously [21]. The work had applied a “chemical model” of C57Bl / 6 mouse Parkinson's disease, which was obtained by subcutaneous injection of 40 mg / kg of dopaminergic proneurotoxin 1-methyl-4-phenyl-1,2,3,6-tetrahydropyridine (MPTP) into these mice [38,39 ]".

2. In Materials and Methods 2.3, you have to describe the cells what you used in here.

A more accurate statement was introduced into line 130: “E. coli сells (strain BL21 (DE3) expressing T7 RNA polymerase”

3. Please provide information on the sample sizes used, including a justification for this number. Where relevant, report the number of independent replications for each experiment. For more details please see here: https://journals.plos.org/plosone/s/submission-guidelines.#loc-statistical-reporting

Thank you for your comment. We have made changes into the main text due to the requirements.

In section 2.9 Quantitative PCR in line 247, a sentence was added: «Each sample was carried out in triplicate».

In section 2.10 we made changes in line 267: «C57BL / 6j mice at the age of 2.5–3 months weighing 25–30 g were used in three groups, N = 5-7 per group (the number of animals per group is sufficient to achieve reliable results). Three isoforms of GDNF, namely cmGDNF, mGDNF1-Coli, mGDNF2-Coli, were administered to each group of animals, respectively».

In section 2.11 clarification on line 282 was added: Control animals were administered with MPTP without a preliminary injection of GDNF (N = 5) (the number of animals per group is sufficient to achieve reliable results).

4. You have to write ‘mL’ instead of ‘ml’, for example, ‘The culture broth of the producers of recombinant mGDNF (750 ml) was cooled…….’ -> ‘The culture broth of the producers of recombinant mGDNF (750 mL) was cooled…..’.

Corrected mL to ml.

5. In Materials and Methods 2.5, you used PC12, E5 rat embryos, SH-SY5Y cells in here, but you did not mentioned the source of cells. Where did you get the cells? If you buy it, you have to mention it. If you get them from other researcher, you have to describe it.

Thank you for your comment. We made changes to paragraph 2.5 in line 206:

“PC12 cells were kindly provided by Institute of Cytology Russian Academy of Science.

E15 rat embryos were isolated by us from E15 embryos of Sprague-Dawley rats.

SH-SY5Y was kindly provided by Dr. E. Yu. Rybalkina from Laboratory of genetics of tumor cells, Research Institute of Carcinogenesis, N.N. Blokhin FSBI NMITs of Oncology, Ministry of Health, Russian Federation. "

6. In Figure 5, you have to stain nuclei with DAPI or others for distinguish.

Unfortunately, we did not find primary samples for nuclei staining, but we did find sharper figures to confirm the accuracy of staining for b3-tubulin of the cell bodies. Figure 5 was loaded as an additional file (Fig5add.tiff, private url at Harvard Dataverse https://dataverse.harvard.edu/privateurl.xhtml?token=df6a30d2-a77b-4829-884b-28cd2e41ddee), where cells with both specific staining and cells with background nonspecific luminescence are clearly visible.

7. In discussion, I want to know what the next is. You have to suggest of the next step to use your technique or the data.

The Discussion some of our plans for further research have been included in line 591.

“Further researches focused on the production of modified GDNF isoforms from mammalian cells and investigation their secretion pathways, as well as their interaction with cell receptors are required. The search for small molecules based on GDNF and able pass the blood-brain barrier and analysis of their neurotrophic potential for clinical use in neurodegenerative diseases are to be continued.”

Reviewer #1: The purpose of this manuscript is to demonstrate the feasibility and usefulness in using molecular biology transfection approach to generate mGDNF protein from HEK293 cells. Despite the demonstration, there is no scientific advance in the approach. The outcome is predictable. For this reason, there is no need to publish such a study.

We appreciate for careful reviewing our manuscript.

We would like to make the goal of this study clearer. As stated in the Introduction Section, glial neurotrophic factor (GDNF) is valuable substance as a therapeutic agent for neurodegenerative diseases, in particular Parkinson's disease. But attempts to translate it into medical practice have failed, despite a success of preclinical trials. Several reasons had been suggested, and as a result of fundamental research, an existence of several GDNF isoforms has been discovered (Lonka-Nevalaita L, Lume M, Leppanen S, Jokitalo E, Peranen J, Saarma M (2010) Characterization of the intracellular localization, processing, and secretion of two neurotrophic factor splice isoforms. J Neurosci 30: 11403-11413). In addition, in our laboratory, it had been found that the expression of engineered mature form of GDNF, without the pre- and pro-regions, does not interfere with protein secretion and does not impair neurotrophic properties, but on the contrary even enhance it (Kust N, Panteleev D, Mertsalov I, et al (2015). Existence of pre- and pro-regions of transgenic GDNF affects an ability to induce axonal sprout growth (Mol Neurobiol 51: 1195-1205 https://doi.org/10.1007/s12035-014-8792-8 )).

Besides, the failure could be either due to problems with a distribution of the protein in the brain after injection, because of its large size (Salvatore MF, Ai Y, Fischer B, et al (2006) Point source concentration of GDNF may explain failure of phase II clinical trial. Exp Neurol 202: 497-505. https://doi.org/10.1016/j.expneurol.2006.07.015), or due to a possible immunogenicity of the protein (Tatarewicz SM, Wei X, Gupta S, et al (2007) Development of a maturing T-cell-mediated immune response in patients with idiopathic Parkinson's disease receiving r-metHuGDNF via continuous intraputaminal infusion. J Clin Immunol 27: 620-627. Https://doi.org/10.1007/s10875-007- 9117-8).

From all mentioned above, the goal of research becomes clearer: it is dedicated to an analysis of activity of various forms of GDNF, as a neurotrophic factor, being produced from different cells. It is known, that usage of E. coli as a producer for mammalian proteins has limitations, in particular, for medical applications. Nevertheless, for several basic and clinical studies, GDNF protein had been produced in prokaryotic systems. 

Therefore, a study of variations in activities of GDNF produced by either mammalian cells (HEK293) or E. coli cells would reveal reasons of success of in vitro results contrary to failure of clinical trials.

These results show how does an activity of GDNF differ, depending on the producing cells (E. coli or mammalian cells), and hence, the protein folding. In addition, for GDNF folding we propose using the recipient serum, which significantly improved neuroinductive properties of the factor.

Reviewer #2: In this research article, the authors describe the effectiveness of various forms of glial-derived neurotrophic factor (GDNF) as a neural protective and regenerative agent. In particular, the manuscript describes differences in activity between GDNF produced by and isolated from E. Coli and mammalian (HEK293) cells, across several in vitro assays and one animal model of Parkinson’s Disease. Overall, their findings suggest that the use of GDNF isoforms expressed by HEK293 cells leads to increased neural morphology in vitro. Furthermore, mice treated with mammalian-expressed GDNF isoforms had increased tyrosine hydroxylase activity (implying increased DOPA production) in the brain, and better motor control compared to E. Coli-expressed GDNF isoforms. Moreover, when additional protein folding steps were incorporated for E.Coli-expressed GDNF isoforms, neuroregenerative capacity more closely resembled that of the mammalian-expressed isoforms. Taken together, these results help shed light on previously unexplained inconsistencies in the effectiveness of GDNF in vitro and in clinical trials, and suggests that with careful engineering, production of therapeutic GDNF using E.coli or other prokaryotic systems may be possible.

Overall, the manuscript is well-written, compelling, and the results conveyed seem to constitute a significant contribution to the field. There are a few areas where additional clarity would help the reader better understand the context and impact of the work, as outlined below.

• In the Introduction, the authors state that “Potential causes [of controversial results in GDNF clinical trials] were discussed by Tenenbaum and Humbert-Claude.” However, these potential causes are not summarized until the Discussion. Moving at least some parts of the Discussion, particularly where the potential causes of controversial clinical trial results are described, would help the reader place the work in context from the beginning and help lay a strong foundational rationale for why the work was performed.

We are grateful to the reviewer for a detailed and careful reading. A lot of comments have delighted us, since the reviewer have examined our study very thoroughly. We have made efforts to answer questions, to accept comments, and to correct the text.

We have tried to expand and clarify the importance of research in the Introduction. As recommended by the reviewer, we have moved a part of the Discussion into the Introduction to line 39:

“For example, Barry Hoffer and later Anders Björklund with colleagues experimentally studied GDNF as an agent restoring dopamine neurons in animal models of Parkinson’s disease [4, 10, 11] and obtained quite promising results. However, during the Phase II of clinical trials no efficiency of recombinant GDNF had been found, and that was the reason of underestimation of possibility of its application for this therapeutic area. Several attempts were made to explain the unsuccessful results. For instance, recombinant GDNF could have abnormal distribution in the cerebral parenchyma due to its heparin-binding properties [12] or could be immunogenic [13]” 

We have made additional comments:

On line 50: “The last approaches are the most perspective and they require further study. The reason of unsuccessful clinical trials could be, for example, a wrong isoform of the protein selected for the therapy, because of diversity of natural GDNF.” 

On line 70: “Besides, it is important to understand whether E. coli is applicable for producing of mGDNF, and what would be properties of the protein after changing the expression system, as well as removal of GFP fusion region for the C-terminal.” 

On line 94: “In addition, the neuroprotective activity of mGDNF (produce from E. coli or HEK293) was studied in the MPTP model of Parkinson’s disease in vivo, the tyrosine hydroxylase activity of substantia nigra pars compacta was assessed, and the motor activities of mice, treated with GDNF isoforms, was compared. The main goal of the research is to understand how important to produce mGDNF in mammalian cells for the keeping mGDNF functioning, or whether it is possible to produce it from E. coli. "

• The authors should more thoroughly describe methods used to count the number of intersections between ganglion neuron processes (Section 2.7), including whether the process was automated.

Thank you for your comment. In Section 2.7. the requested information has been added.

“The activity of the produced neurotrophic factors was quantified by counting the processes of spinal ganglion neurons stained with antibodies against β3-tubulin. The number of crosses was counted for the processes with diameters of 500 and 750 µm using “Multi-point” tool in ImageJ program.”

• In Figure 4e, it seems as though there are a greater number of intersections in the smaller diameter counting circle than in the larger one. The authors should explain why this might be the case. (This discrepancy may also be explained by more clear methods in Section 2.7, as suggested in the previous point).

In reference to the Figure 4e an explanation was added to line 406:

“After adding cmGDNF, there is an equal number of intersections on the small and large diameter counting circles. On the contrary, adding other forms of GDNF have yielded more intersections on the small circle relative to the large one. Perhaps due to activity of factors, the fibers begin to branch earlier and do not grow to a large circle. "

• Commercially-sourced GDNF (“recGDNF”) was used as a control in several experiments. Based on the information provided (e.g., source: Peprotech), this GDNF also appears to have been produced by E.Coli. The authors should describe, to the extent possible (understanding that some of Peprotech’s information may be proprietary), how this protein differs from those produced in-house for the study.

There is an insert into Section 2.4. to line 178:

“Recombinant mGDNF /Coli (produced in-house) was taken as internal control of protein production under the same conditions, recombinant commercial recGDNF (Peprotech, USA) was chosen as an external control, as a conventional control accepted worldwide for studies of neutrophic properties of GDNF".

• In reference to Figure 4, the authors state that “…exposure to cmGDNF from HEK293 cells induced a smaller number of processes but they were longer; and only cmGDNF induced the formation of longer processes.” The discussion should include some interpretation of these results, including whether this property is more or less beneficial than numbers of intersections, and in what clinical scenarios length vs. branching, or intersections, may be most relevant.

Thanks for the comment, some interpretations were included into the Discussion to line 543:

“The fact that only cmGDNF induced the formation of longer processes compared to other studied GDNF isoforms deserves special attention and suggests the possibility of using the isoforms of GDNF in different clinical cases/nosology. After peripheral nerve injuries, it is important that the regenerating axons reach their target as quickly as possible [44]. In this regard, the effect of cmGDNF may represent a promising tool for the damaged nerves recovery therapy [44, 45]. In case of a violation of local connections between neurons, for example, during ischemia, secondary progressive multiple sclerosis, or neurodegenerative diseases, rapid recovery of the network is required [46-48], which can be provided by stimulating the formation of the neuronal short and branched processes.”

• Keeping the order of experimental groups consistent between all figures in the manuscript, and/or adding in-figure text labels for each panel, would be helpful for the reader.

Thanks for the comment, text labels have been added to the Figures 1,3,4,5. In Figures 2, 6, 7, 8 all necessary text labels have already been marked. This has updated Figures 1, 3, 4, and 5.

• The authors state that all data are contained within the manuscript and supporting information, but individual data points and/or additional images are not included. These should be made available via a public repository or similar, according to PLOS One guidelines.

All original image data are available from the Harvard Dataverse Network database (accession number https://doi.org/10.7910/DVN/DIYBUS) (https://dataverse.harvard.edu/privateurl.xhtml?token=a01509d7-a3a6-41c5-a646-de9ab3a13dbc)

• In reference to Figure 4, the authors state that “The results…were equally as amazing…” This should be rephrased so as to be objective and technical (syntax).

Thanks for the comment, it was rephrased with the word ‘unforeseen’ in line 393:

“The results of the exposure of spinal ganglion culture to Pro-mGDNF /coli were equally unforeseen as of PC12 cells.”

The Abstract has been slightly modified:

Abstract: The glial cell line‐derived neurotrophic factor, GDNF, is involved in a survival of dopaminergic neurons. Besides, GDNF can induce axonal growth and creation of new functional synapses. GDNF potential is promising for translation to treat diseases associated with neuronal death: neurodegenerative disorders, ischemic stroke, and cerebral or spinal cord damages. Unproductive clinical trials of GDNF for Parkinson’s disease treatment have induced to study this failure. A reason could be due to irrelevant producer cells that can not perform the required post-translational modifications. The cellular activity of recombinant mGDNF produced by E. coli have been compared with mGDNF produced by human cells HEK293. mGDNF variants were tested with PC12 cells, rat embryonic spinal ganglion cells, and SH-SY5Y human neuroblastoma cells, in vitro as well as with a mouse model of the Parkinson’s disease in vivo. Both in vitro and in vivo the best neuro-inductive ability belongs to mGDNF produced by HEK293 cells.

In Section 2.10 of Methods and Materials on lines 274 and 275, the words ‘recombinant mGDNF’ were replaced with ‘isoforms GDNF’.

In the Results section:

• The sentence has been added to line 315: “In the Table 1 the names of the growth factors and the sources of proteins were mentioned”.

• A notice has been added to line 339 in the Legend to the Figure 2. "Arrows indicate expressed proteins."

The Discussion section has also been expanded:

• Added on line 514:

“The problem of the inexpedient application of E. coli for the production of recombinant proteins has been discussed for some time in various publications. It has already been shown that physiological properties of E. coli limit their use for the production of proteins in their native form, especially polypeptides that are subjected to major post-translational modifications [44]. Recent articles assess the change in the properties of specific proteins produced by different producer cells. For example, in the work of Yu et al [45], such an analysis carried out for Recombinant human IFNα2b (rhIFNα2b) synthesized by various producers demonstrated the inexpediency of using yeast cells.”

• Added on line 552:

“…which suggests the need of a more thorough analysis of GDNF isoforms as neuroinducers, since a wrong choice of an isoform can lead to a neural differentiation blockade..”

• Added on line 557:

“The study of the ability of the obtained factors to maintain the viability of cells in vitro on the human neural cell culture SH-SY5Y demonstrated low activity of mGDNF and Pro-mGDNF. 

We can even state that Pro-mGDNF (GDNF with the deleted Prе-region) showed more significant results as a factor supporting the viability of SH-SY5Y neuroblastoma cells. 

Besides, the investigation of the neuroinductive properties of recombinant mGDNF and Pro-mGDNF on the SH-SY5Y neuroblastoma cell culture showed an insignificant efficacy of Pro-mGDNF and a more significant effect of mGDNF. The results demonstrate that mGDNF isolated from E. coli loses the ability to maintain cell viability, but has good neuroinductive properties in vitro. 

In addition, the analysis of mGDNF and Pro-mGDNF compared to the recombinant recGDNF protein showed that the results obtained for factors with a deleted pro-region are not characteristic for "normal" GDNF with the pro-region of normal length. It should be noted that this protein is secreted through the Golgi apparatus and may have properties other than the factors obtained in the current study. 

Thus, the results obtained confirm our hypothesis that the GDNFs secreted in the body in different ways (through the Golgi apparatus or directly through the vesicles) possess different properties. Probably, GDNF with a long pro-region is required to maintain cell viability, while GDNF with a shortened pro-region (or, in our case, with a deleted one) may stimulate neural differentiation of immature cells. Thus, mGDNF can be applied for neural cells recovery after their traumatic death or in case of neurodegenerative diseases.”

• Added on line 577: 

“Firstly, this indicates the imperfection of the in vitro models for the analysis of neuroinductive properties of different molecules. However, this may seem acceptable, because the evaluation of the neuroinductive properties in brain tissue where cells have a different density of interaction with each other is of great importance. Despite the fact that the in vitro model is still used due to its simplicity, the conclusions about the effectiveness of a neuroinductive molecule can only be drawn after the consistent in vitro and in vivo studies. Secondly, the results of the current work once again confirm the importance of using mammalian cells as GDNF producers. 

We would also like to note our undoubted success in obtaining a more effective GDNF neuroinducer molecule from E. coli using the recipient animal serum. This may indicate the absence of the formation of the necessary bonds (for example, disulfide bonds) in the tertiary structure of the protein in the case of standard refolding.”

We have added new links to citations in the References according to the corrections in the text.

---

## [Editor Report · Decision Letter 1]

24 Sep 2021

Neuroinductive properties of mGDNF depend on the producer, E.coli or human cells

PONE-D-20-31228R1

Dear Dr.Shamadykova,

We’re pleased to inform you that your manuscript has been judged scientifically suitable for publication and will be formally accepted for publication once it meets all outstanding technical requirements.

Kind regards,

Sujeong Jang

Academic Editor

PLOS ONE
---

## [Editor Report · Acceptance letter]

30 Sep 2021

PONE-D-20-31228R1 

Neuroinductive properties of mGDNF  depend on the producer, *E. Coli* or human cells 

Dear Dr. Shamadykova:

I'm pleased to inform you that your manuscript has been deemed suitable for publication in PLOS ONE. Congratulations! Your manuscript is now with our production department. 

Kind regards, 

on behalf of

Dr. Sujeong Jang 

Academic Editor

PLOS ONE